# Xanthine Oxidase Inhibitor Febuxostat Exerts an Anti-Inflammatory Action and Protects against Diabetic Nephropathy Development in KK-Ay Obese Diabetic Mice

**DOI:** 10.3390/ijms20194680

**Published:** 2019-09-21

**Authors:** Yu Mizuno, Takeshi Yamamotoya, Yusuke Nakatsu, Koji Ueda, Yasuka Matsunaga, Masa-Ki Inoue, Hideyuki Sakoda, Midori Fujishiro, Hiraku Ono, Takako Kikuchi, Masahiro Takahashi, Kenichi Morii, Kensuke Sasaki, Takao Masaki, Tomoichiro Asano, Akifumi Kushiyama

**Affiliations:** 1Department of Medical Science, Graduate School of Medicine, University of Hiroshima, 1-2-3 Kasumi, Minami-ku, Hiroshima City, Hiroshima 734-8551, Japan; d186723@hiroshima-u.ac.jp (Y.M.); ymmty@hiroshima-u.ac.jp (T.Y.); nakatsu@hiroshima-u.ac.jp (Y.N.); urouedakouji@yahoo.co.jp (K.U.); ymatsunaga@tulane.edu (Y.M.); b131831@hiroshima-u.ac.jp (M.-K.I.); 2Center for Translational Research in Infection & Inflammation, School of Medicine, Tulane University, 6823 St. Charles Avenue, New Orleans, LA 70118, USA; 3Division of Neurology, Respirology, Endocrinology, and Metabolism, Department of Internal Medicine, Faculty of Medicine, University of Miyazaki, 5200 Kihara, Kiyotake, Miyazaki 889-1692, Japan; hideyuki_sakoda@med.miyazaki-u.ac.jp; 4Division of Diabetes and Metabolic Diseases, Nihon University School of Medicine, Itabashi, Tokyo 173-8610, Japan; fujishiro.midori@nihon-u.ac.jp; 5Department of Clinical Cell Biology, Graduate School of Medicine, Chiba University, 1-8-1 Inohana, Chuo-ku, Chiba City, Chiba 260-8670, Japan; hono@chiba-u.jp; 6Division of Diabetes and Metabolism, The Institute for Adult Diseases, Asahi Life Foundation, 2-2-6, Nihonbashi Bakurocho, Chuo-ku, Tokyo 103-0002, Japan; kikuchi-tk@umin.ac.jp; 7Department of Pharmacotherapy, Meiji Pharmaceutical University, 2-522-1 Noshio, Kiyose City, Tokyo 204-8588, Japan; t-masa@my-pharm.ac.jp; 8Department of Nephrology, Hiroshima University Hospital, 1-2-3 Kasumi, Minami-ku, Hiroshima City, Hiroshima 734-8551, Japan; kenichi_morii@yahoo.co.jp (K.M.); sasakikuma@gmail.com (K.S.); masakit@hiroshima-u.ac.jp (T.M.)

**Keywords:** diabetic kidney diseases, xanthine oxidase, glomerular damage

## Abstract

Hyperuricemia has been recognized as a risk factor for insulin resistance as well as one of the factors leading to diabetic kidney disease (DKD). Since DKD is the most common cause of end-stage renal disease, we investigated whether febuxostat, a xanthine oxidase (XO) inhibitor, exerts a protective effect against the development of DKD. We used KK-Ay mice, an established obese diabetic rodent model. Eight-week-old KK-Ay mice were provided drinking water with or without febuxostat (15 μg/mL) for 12 weeks and then subjected to experimentation. Urine albumin secretion and degrees of glomerular injury judged by microscopic observations were markedly higher in KK-Ay than in control lean mice. These elevations were significantly normalized by febuxostat treatment. On the other hand, body weights and high serum glucose concentrations and glycated albumin levels of KK-Ay mice were not affected by febuxostat treatment, despite glucose tolerance and insulin tolerance tests having revealed febuxostat significantly improved insulin sensitivity and glucose tolerance. Interestingly, the IL-1β, IL-6, MCP-1, and ICAM-1 mRNA levels, which were increased in KK-Ay mouse kidneys as compared with normal controls, were suppressed by febuxostat administration. These data indicate a protective effect of XO inhibitors against the development of DKD, and the underlying mechanism likely involves inflammation suppression which is independent of hyperglycemia amelioration.

## 1. Introduction

Diabetic kidney disease (DKD) is currently a leading cause of end-stage renal failure [1], while treatments with renin–angiotensin–aldosterone system blockers with normalization of hyperglycemia are regarded as the gold standard [2,3]. In addition, several studies have shown hyperuricemia to be an independent factor impacting insulin resistance and also to play a significant role in the development of diabetic nephropathy [4,5]. In patients with type 1 diabetic nephropathy, usually not accompanied by insulin resistance, hyperuricemia is also shown to be an independent risk of renal dysfunction [6].

Uric acid is generated from hypoxanthine and xanthine by xanthine oxidase (XO) as the final step in the metabolism of endogenous and exogenous purines [7]. Clinical trials have proven the efficacy of XO inhibitors, including allopurinol and febuxostat, not only for lowering the serum uric acid concentration but also for protection against the progression of renal diseases [8,9]. Even for diabetic nephropathy, XO inhibition has been shown to be effective in several clinical trials [10,11,12]. However, whether reducing the serum uric acid level or suppression of XO activity contributes to the protective effect of XO inhibitors on renal diseases remains unclear. Therefore, elucidation of the effects and mechanisms of action of XO inhibitors is eagerly awaited.

A previous study using an insulin-deficient Type 1 diabetic model rat and XO inhibitors, including allopurinol and febuxostat, suggested the renal protective effect of febuxostat to be mediated via attenuation of oxidative and inflammatory effects [13,14]. On the other hand, a study using obese diabetic db/db mice treated with allopurinol also showed the renal protective effect of XO inhibitors to involve lowering uric acid directly [13,15]. Therefore, to date, only a few studies have focused on the mechanism, particularly that against diabetic nephropathy, underlying XO inhibitor-mediated renal protective effects. 

In this study, we employed KK-Ay mice that spontaneously exhibit type 2 diabetes associated with hyperglycemia, glucose intolerance, hyperinsulinemia, obesity, and microalbuminuria [16], and investigated the effects of febuxostat on the development of diabetic nephropathy in these mice. Our observations raise the possibility that the renal protective effects of the XO inhibitor febuxostat are mediated mainly by an anti-inflammatory effect, which is independent of the effects on glycemic control.

## 2. Results

### 2.1. Effects of XO Inhibitor Febuxostat on Glycemic Control and Serum Uric Acid Levels

Diabetic KK-Ay mice were divided into two groups, with and without febuxostat in drinking water. After 12 weeks of treatment, while KK-Ay mice had higher body weights, blood glucose concentrations in the fed state, serum glycated albumin, and serum uric acid levels than the wild-type mice (Figure 1A–D), febuxostat treatment reduced the serum uric acid level of KK-Ay mice without significantly affecting other parameters. However, unexpectedly, glucose tolerance and insulin tolerance tests revealed treatment with febuxostat significantly ameliorated the impairments in KK-Ay mice (Figure 1E,F). 

### 2.2. Effects of Febuxostat Administration on Glomerular Sclerosis

Glomerular sclerosis is among the features of diabetic nephropathy. The degrees of glomerular sclerosis were estimated based on the findings of Hematoxylin–Eosin (HE) staining and positive peroxide acid-Schiff (PAS) staining (Figure 2A,B). PAS staining is reportedly useful for definitively demonstrating glomerular hypertrophy, an early feature of diabetic nephropathy [17]. Both HE and PAS staining showed glomerular sclerosis to be markedly advanced in KK-Ay mice as compared with wild-type mice, and that this progression showed significant attenuation in response to febuxostat treatment in KK-Ay mice (Figure 2B). Glomerular injury score (GIS) and glomerular areas calculated based on PAS-stained areas were markedly increased in KK-Ay mice than in wild-type controls, and these changes were partially but significantly normalized in febuxostat-treated KK-Ay mice (Figure 2C). While febuxostat treatment did not normalize the increased kidney weights of KK-Ay mice as compared with those of wild-type mice (Figure 2D), an approximately 10-fold increase in the urinary albumin-to-creatinine ratio (ACR) in KK-Ay mice was markedly ameliorated, to a level near that of control C57BL/6 mice, by febuxostat treatment (Figure 2E).

### 2.3. Effects of Febuxostat on the Renal Expression Levels of Inflammatory Cytokines

Since elevated inflammation-related cytokine and chemokine expressions are reportedly involved in the pathogenesis in diabetic nephropathy, the effects of febuxostat on their mRNA levels were evaluated. In comparison with the normal mice, renal mRNA levels of IL-1β, IL-6, MCP-1, CXCL1, CXCL2, and CXCL5, but not those of the macrophage marker F4/80 and TNFα, were markedly elevated in the KK-Ay mice (Figure 3). Treatment with febuxostat essentially normalized the upregulated expressions of inflammatory cytokines (IL-1β, IL-6) without significantly affecting, though a tendency for reduction was observed, inflammatory chemokines (CKCL1, CXCL2, CXCL5) in KK-Ay mouse kidneys (Figure 3).

### 2.4. Effects of Febuxostat on Fibrosis-Related Collagen Gene Expressions 

Azan staining was performed to evaluate the degree of fibrotic change, a change typical of the advanced stage of diabetic nephropathy. Imaging analysis of the fibrotic area based on Azan staining showed that the areas of increased fibrotic staining and fibrogenesis including glomerulosclerosis in the KK-Ay mice, as compared with control mice, were normalized by febuxostat treatment (Figure 4A). We next investigated the effects of febuxostat on the expression levels of the mRNA of genes related to the fibrotic process. The mRNA expression level of collagen 1a1 was increased in the kidneys of KK-Ay mice, but this elevation tended to be normalized by febuxostat treatment though the difference did not reach statistical significance (Figure 4B). On the other hand, mRNA expression levels of collagen 1a2, collagen 4a1, and collagen 4a2 did not show significant differences between KK-Ay and wild-type mice or between the presence and absence of febuxostat treatment (Figure 4B).

### 2.5. Effects of Febuxostat on Markers of Oxidative Stress and the Endoplasmic Reticulum

Since febuxostat reportedly reduces the production of xanthine-induced oxidants, we measured oxidative stress in the kidneys of KK-Ay mice with and without febuxostat treatment and in wild-type mice. The amount of renal MDA, a product of lipid peroxidation, measured by the thiobarbituric acid reactive substances (TBARS) assay, showed no significant differences with wide variations among the three groups (Figure 5A). Expressions of endothelial damage markers (ICAM-1, VCAM-1) in the kidneys were elevated in KK-Ay mice as compared with wild-type mice, and febuxostat reduced these ICAM-1 elevations though not significantly (Figure 5B). Similarly, the mRNA levels of an endoplasmic reticulum stress marker (CHOP) were elevated in KK-Ay mice as compared to wild-type mice, with febuxostat reducing, though not significantly, these elevations (Figure 5C).

## 3. Discussion

The relationship between hyperuricemia and the development of DKD is now widely recognized [18,19,20]. In the pathogenesis of DKD, inflammation appears to play essential roles via inflammatory mediators, adhesion molecules, and inflammatory signaling pathways directly or indirectly induced by hyperglycemia [1]. We previously demonstrated XO activity to promote inflammation in blood vessels [21] and the liver [22], while XO inhibitors suppress inflammation and oxidative stress [23] in macrophages. Thus, treatment with XO inhibitors prevented inflammatory and fibrotic changes in rodent models of atherosclerosis and non-alcoholic steatohepatitis. Another group also reported febuxostat to suppress angiotensin II-induced aortic fibrosis via XO inhibition in macrophages [24].

In this study, we showed febuxostat treatment to attenuate the development of proteinuria and to ameliorate mesangial structural changes in the kidneys of type 2 diabetic KK-Ay mice. A previous study using db/db mice also showed that allopurinol normalized hyperuricemia (to approximately 3 mg/dL) and reduced albuminuria with improvements in some of the features of diabetic nephropathy [15]. The authors insisted that high glucose accompanied by high uric acid (UA) is the causal mechanism of progressive renal function loss in db/db mice, based on a small increase in sICAM-1 detected by in vitro assay. In contrast, while KK/Ay mice showed slightly higher serum UA levels than control wild-type mice, these levels remained relatively low (around 1 mg/dL). The improved glucose tolerance and insulin tolerance test results by febuxostat obtained in this study might be consistent with the previous study using XO inhibitor [25]. However, it should be noted that the actual degree of hyperglycemia in KK/Ay mice, judging from blood sugar and glycated albumin levels, was not significantly altered by febuxostat administration. Furthermore, we recently reported treatment with febuxostat to provide significant protection from kidney failure, possibly via suppression of inflammation, in gddY mice [26], a model spontaneously developing IgA nephropathy [27] but showing no glucose metabolism defect. Taken together, these findings suggest that it would be inappropriate to interpret the mechanism underlying amelioration of renal dysfunction in KK/Ay mice as being via normalization of hyperuricemia and hyperglycemia.

The renal protective effects of febuxostat appear to be exerted through its potent anti-inflammatory actions. Despite macrophage infiltration, as indirectly evaluated by the F4/80 level, being unchanged by febuxostat treatment, renal injury was suppressed, and this was associated with reductions in inflammatory cytokines. XO inhibition was also reported to attenuate certain cellular migrations [28,29] and monocytic differentiations, such as those of foam cell formation and multinucleated giant cell formation [21,30]. Indeed, the renal protective effects of XO inhibition are consistent with previous reports showing the effects of febuxostat in mice with gddY IgA nephropathy [26] and in 5/6 nephrectomy rats, regardless of the coexisting hyperuricemia [31]. The suppressive effect on inflammation is a common feature of febuxostat treatment. Our present observations suggest febuxostat suppresses urinary albumin and glomerular injury at early stages of type 2 diabetic nephropathy, i.e., at the stage when inflammatory cytokines, such as IL-6 [32] and MCP-1, are secreted [33]. Repeated suppressions of ICAM-1 expression reportedly raised the possibility of preventing tissue injuries by administering XO inhibitors [15,34,35].

Collagenous proliferation evaluated by Azan as well as mesangial expansion, as evaluated by PAS staining, became more severe in KK/Ay mice and showed amelioration in response to febuxostat. In db/db, there are reportedly no changes in PAS staining with allopurinol treatment [15]. Febuxostat attenuated renal protein expression of TGF-β, CTGF, and collagen 4 in the kidneys of Zucker obese rats [14]. Studies using db/db mice [15] and our current experiments showed no reductions in collagen 4, though our data on collagen 1a1 and measurement of collagen 3 in db/db mice indicated both to be reduced by febuxostat administration. Despite variations in renal glomerular damage among animal models and/or species, it is reasonable to consider febuxostat to exert positive effects in terms of maintaining glomerular structure. Comparisons of the data from different sources imply that the effect of febuxostat might be more evident in KK/Ay than in db/db mice, although the mechanism underlying this difference is unclear.

Two mechanisms possibly underlying hyperuricemia-related DKD development have been suggested, elevation of the serum UA and increased superoxide free radical generation [36,37]. Oxidative stress indicators, such as the whole-kidney MDA level, and ER stress marker CHOP did not show a clearly significant difference and relatively large variance was noted. We speculate that the generation of oxidative stress would be limited to a highly localized area and that significant local changes might be difficult to detect at the whole-kidney level. Thus, we were not able to conclude that the effects of febuxostat are not due, even partially, to reduced free radical generation. Further studies are necessary to clarify this issue.

There are some limitations in this study. First, since febuxostat administration to kk-ay mice in this study showed significant changes in glomerular lesions and no obvious changes in the renal tubules or mesenchyme. Therefore, we focused on the histological glomerular change, however, the inflammation and/or fibrosis in tubulointerstitium should be evaluated in the future study. Finally, all the mRNA analysis is performed in whole kidney and not from the glomeruli, therefore the relationship between the gene expression and the effect of febuxostat on the glomerular protection might not be direct. Finally, the other xanthine oxidase inhibitors such as allopurinol, its metabolite oxypurinol, and topiroxostat should be evaluated in our model, to clarify the glomerular protective effect of febuxostat is class-effect.

In conclusion, we demonstrated that KK-Ay mice developed evident hyperglycemia and slight hyperuricemia, leading to diabetic nephropathy progression. XO inhibition by febuxostat ameliorated both proteinuria and glomerular damage, without affecting plasma glucose levels. Furthermore, the mechanism likely involves blocking intrarenal inflammation.

## 4. Materials and Methods

### 4.1. Animals

Male diabetic KK-Ay mice and age-matched C57BL/6 mice (8 weeks of age) were purchased from CLEA Japan (Tokyo, Japan). Febuxostat was obtained from Teijin Pharma Ltd., Tokyo, Japan. The mice were housed in temperature- and light-controlled rooms with free access to food (Oriental Yeast, Tokyo, Japan) and water. After the 12 weeks of treatment with or without drinking water containing febuxostat (15 μg/mL), which corresponds to approximately 1 mg/kg body weight per day, close to the clinical dose range of oral febuxostat, the mice were killed, and their kidneys and blood samples were collected. For uric acid (UA) measurement, 100 μL whole blood samples were collected with 100 μM allopurinol (Wako, Osaka, Japan). To measure the glucose concentration, blood samples were incubated on ice for 30 min and then centrifuged at 15,000 rpm for 30 min at 4 °C. All samples were preserved at −80 °C. All mice were handled in accordance with Guidelines for Care and Use of Experimental Animals published by Hiroshima University, and all protocols were approved by Hiroshima University (Approval #A17-178 Date 29 March 2018).

### 4.2. Metabolic Analysis

Mouse urine was collected for 24 h employing metabolic cages and preserved at −80 °C. Urine albumin and creatinine were quantified using a mouse albumin ELISA kit (FUJIFILM Wako Shibayagi, Gunma, Japan) and a mouse creatinine assay kit (Abcam, Cambridge, MA, USA), respectively. Serum UA concentrations were assayed with a UA assay kit (Cayman Chemical, Ann Arbor, UI, USA). All assays were conducted according to the manufacturers′ protocols. The albumin-to-creatinine ratio (ACR) was calculated as urinary albumin/urinary creatinine.

### 4.3. Histological Study

Kidneys were fixed in 10% formalin, embedded in paraffin, and then cut into 2-μm sections. The sections were subjected to peroxide acid-Schiff (PAS) and Azan staining to identify glomerular sclerosis and fibrotic change, respectively [38].

Glomerular injury included mesangial matrix expansion and/or hyalinosis with focal adhesions, capillary dilation, and true glomerular tuft occlusion, and sclerosis. For the quantitative analysis of glomerular injury, the glomerular injury score (GIS) was determined on PAS-stained paraffin sections using a scale ranging from 0 to 4 for normal (0), with 1 = 25% sclerosis, 2 = 50% sclerosis, 3 = 75% sclerosis, and 4 = 100% sclerosis [39]. On average, 40 glomeruli were evaluated per mouse. GIS was calculated as [∑ (each score × number of glomeruli)]/number of glomeruli. The glomerular area was measured in an average of 40 glomeruli per mouse employing image J (NIH, Bethesda, MD, USA). The renal fibrotic area was assessed using Azan staining, with an average of 10 glomeruli being measured per mouse using NIH Image J. All structural analyses were conducted in a blinded manner on unidentified sections.

### 4.4. Malondialdehyde (MDA) assay

Thiobarbituric acid reactive substances (TBARS) were measured using a TBARS assay kit (Cayman Chemical, Ann Arbor, UI, USA), according to the manufacturer′s instructions, and absorbance was determined at 540 nm. 

### 4.5. Quantitative Real-Time RT-PCR

Total RNA was extracted from mouse kidneys using Sepasol reagent (Nacalai Tesque, Kyoto, Japan). A 500 ng quantity of RNA was reverse transcribed using the Verso cDNA Synthesis kit (Thermo Scientific, Vilnius, Lithuania), which degrades double-stranded DNA. Quantitative real-time RT-PCR was performed using SYBR Green PCR master mix (Agilent Technologies, Santa Clara, CA, USA) on a CFX96 real-time PCR system (Bio-Rad, Hercules, CA, USA). Relative mRNA gene levels were normalized to the GAPDH mRNA level, and relative expressions were determined by the comparative Ct method. The designed primers used are shown previously [26].

### 4.6. Statistical Analysis

All results are expressed as means ± SE. Statistical analyses were conducted using ANOVA followed by Dunnett′s test. A value of *p* < 0.05 was taken to indicate a statistically significant difference.

## Figures and Tables

**Figure 1 ijms-20-04680-f001:**
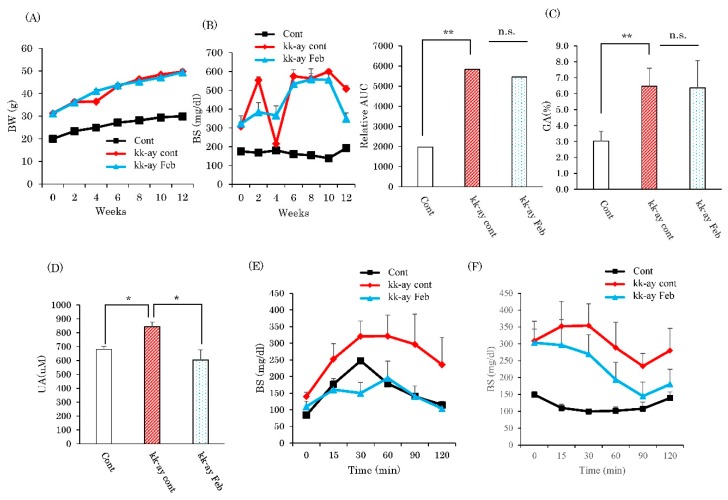
Inhibitory effects of xanthine oxidase inhibitor febuxostat on the progression of diabetic nephropathy. (**A**) Body weights and (**B**) blood sugar levels were measured. (**C**) Serum glycated albumin levels. (**D**) Serum uric acid levels. (**E**) Glucose tolerance test results. (**F**) Insulin tolerance test. Data are means ± SE. * *p* < 0.05, ** *p* < 0.001, n = 8 for control group, n = 10 for KK-Ay mice without febuxostat group and n = 8 for KK-Ay mice with febuxostat group.

**Figure 2 ijms-20-04680-f002:**
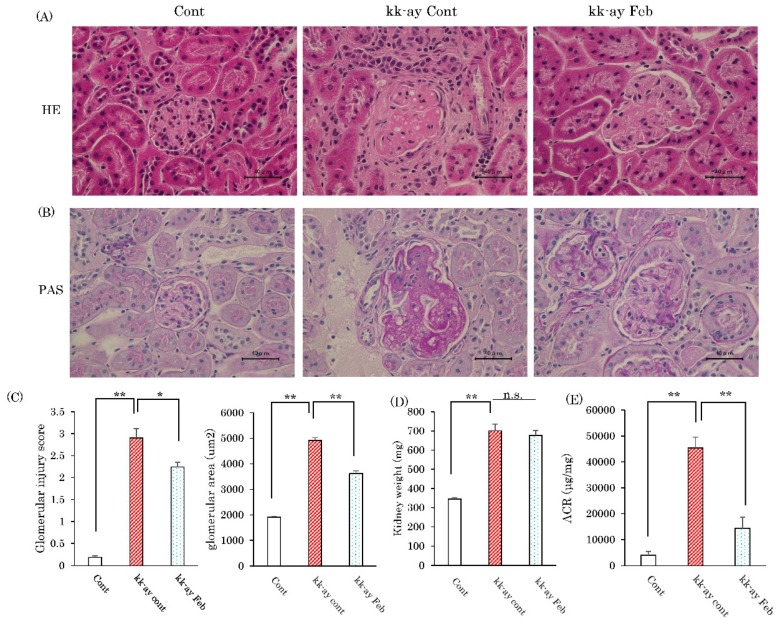
Febuxostat administered for prevention of glomerular sclerosis. (**A**) Hematoxylin–Eosin (HE) staining of kidney fractions. Scale bar = 40 μm. (**B**) Peroxide acid-Schiff (PAS) staining of kidney fractions. Scale bar = 40 μm. (**C**) Glomerular sclerosis score and glomerular area were calculated using image J. (**D**) Kidney weights and (**E**) the urinary albumin to creatinine ratio was measured at 20 weeks. Data are means ± SE. * *p* < 0.05, ** *p* < 0.001. n = 8 for control group, n = 10 for KK-Ay mice without febuxostat group and n = 8 for KK-Ay mice with febuxostat group.

**Figure 3 ijms-20-04680-f003:**
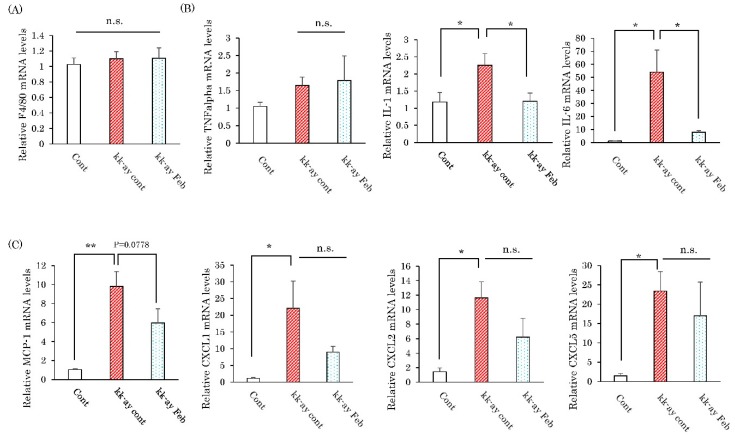
Febuxostat suppressed expressions of inflammatory cytokines in the kidneys of KK-Ay mice. (**A**–**C**) Relative mRNA levels of macrophage markers, cytokines, and chemokines in the kidneys were determined by employing real-time PCR. Data are means ± SE. * *p* < 0.05, ** *p* < 0.001. n = 8 for control group, n = 10 for KK-Ay mice without febuxostat group and n = 8 for KK-Ay mice with febuxostat group.

**Figure 4 ijms-20-04680-f004:**
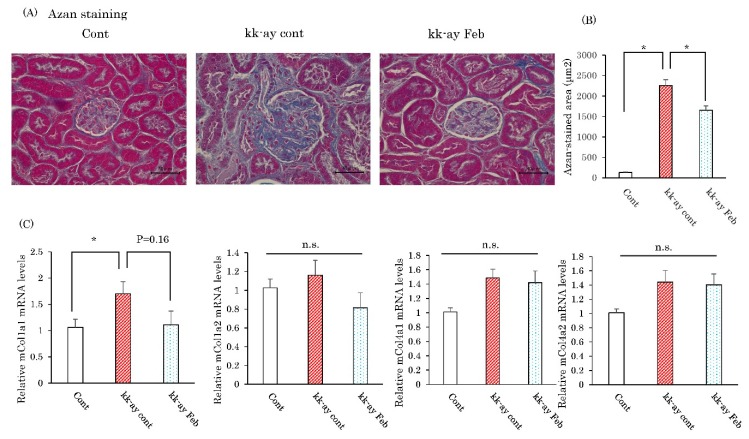
Febuxostat prevented progression of kidney fibrosis. (**A**) Azan staining of kidney fractions. Scale bar = 40 μm. (**B**) Relative mRNA levels of fibrotic markers in the kidneys. (**C**) The renal fibrotic area was assessed using Azan staining and then measured using image J. Data are means ± SE. * *p* < 0.05. n = 8 for control group, n = 10 for KK-Ay mice without febuxostat group and n = 8 for KK-Ay mice with febuxostat group.

**Figure 5 ijms-20-04680-f005:**
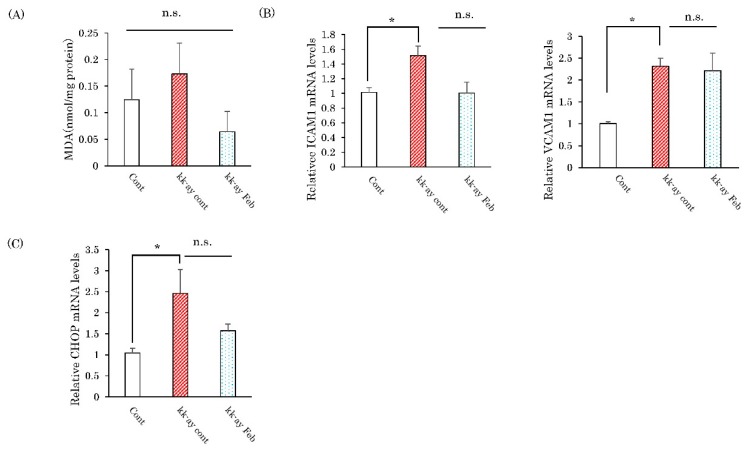
Febuxostat had no significant effect on either oxidative stress or endoplasmic reticulum stress. (**A**) Amounts of malondialdehyde were measured in renal tissues. (**B**) Relative mRNA levels of ICAM-1 and VCAM-1 in the kidneys. (**C**) Relative mRNA levels of CHOP in the kidney. Data are means ± SE. * *p* < 0.05, n = 8 in control group and KK-Ay mice with febuxostat, n = 10 in KK-Ay mice without febuxostat

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
