# Peer review of "Xanthine Oxidase Inhibitor Febuxostat Exerts an Anti-Inflammatory Action and Protects against Diabetic Nephropathy Development in KK-Ay Obese Diabetic Mice"

_ijms, 2019, doi:10.3390/ijms20194680_

Round 1

Reviewer 1 Report

Yu Mizuno et. al, were evaluated the protective effect of febuxostat, a xanthine oxidase inhibitor on proteinuria in KK-Ay obese diabetic mice model. Authors have treated the animals for 12 weeks and measured the body weights, blood sugars, GTT, GIS, glomerular sclerosis, fibrosis and inflammatory cytokine levels.  The authors proposed that xanthine oxidase inhibition by febuxostat improves both proteinuria and glomerular damage by blocking intrarenal inflammation, without affecting plasma glucose levels. Authors are focused the studies only on few markers but not on their role and mechanism. Design of the study could be improved to achieve the aim mentioned in manuscript. Overall, studies mentioned in the manuscript may be beneficial to researchers especially in field of diabetes and renal disorders.

Some imperfections listed below

Line 89: Regents? Suggested to check the spelling and modify it to Methods. Line 203 and figure legend (Fig.5) it is mentioned GHOP. Is it CHOP (C/EBP)? In discussion part It is suggested to mention about the possible reason, how the treatment did not improve the ER stress? Since the xanthine oxidase inhibitors, especially Febuxostat known to improve the ER stress Scale bars are missing in Fig.2A PAS control and all panels of figure 4A 4C labelling is missing. Figure legend 5C mentioned about two markers BiP and GHOP (CHOP) but the image is missing. Authors are suggested to modify the ‘X’ axis tick marks with an angle or with abbreviations in all graphs to make a clear presentation

Author Response

We thank the reviewer 1 for helpful comments.

>Line 89: Regents?

We modify the subtitles of Materials and Methods section. “Regents” is Reagents, and all the reagents are indicated in that section elsewhere.

>Suggested to check the spelling and modify it to Methods. Line 203 and figure legend (Fig.5) it is mentioned GHOP. Is it CHOP (C/EBP)?

Yes, and we did it.

>In discussion part It is suggested to mention about the possible reason, how the treatment did not improve the ER stress? Since the xanthine oxidase inhibitors, especially Febuxostat known to improve the ER stress

We added the sentences below in the discussion section.Line 266.

“Oxidative stress indicators, such as the whole-kidney MDA level, and ER stress marker CHOP did not show a clearly significant difference and relatively large variance was noted.“

We found partial improvement of CHOP elevation in kk-ay (not significant), as mentioned in the result section, therefore we cannot present as the reviewer indicated, “the febuxostat treatment did not improve the ER stress”. However, we think the contribution of ER stress improvement for the treatment of febuxostat is seemed to be small because other changes such as fibrosis are more evident than the ER stress change.

>Scale bars are missing in Fig.2A PAS control and all panels of figure 4A 4C labelling is missing.

We modified all the figures, with scale bars and panel labeling. We also modified P value about Figure4C Co1a1 expression

>Figure legend 5C mentioned about two markers BiP and GHOP (CHOP) but the image is missing.

We are sorry to inadequate description. Figure 5C presented only CHOP expression. Indeed, BiP or other markers for ER stress did not have significant signs or tendency, we speculate there are some feedback for the ER stress responses and varied by some individual state. These are not our focus of the study, and contributions seem relatively small in our investigation.

>Authors are suggested to modify the ‘X’ axis tick marks with an angle or with abbreviations in all graphs to make a clear presentation

We modified all the marks and abbreviations in all graphs.

Reviewer 2 Report

Manuscript by Mizuno et. Al., entitled “Xanthine oxidase inhibitor febuxostat exerts an anti-inflammatory action and protects against diabetic nephropathy development in KK-Ay obese diabetic mice” is an interesting article where authors investigated the role of febuxostat, a xanthine oxidase (XO) inhibitor, in the development of DKD. Treatment with febuxostat (15μg/mL) significantly reduced the urine albumin secretion and glomerular pathological scoring as compared to control. Besides that, febuxostat also significantly improve insulin sensitivity and glucose tolerance. These finding further support the role of XO inhibitor as a protective factor in development of DKD. Although authors performed experiments meticulously and manuscript is nicely written, I have some concern/comments that written below.

Author should measure the urine albumin creatinine ratio (UACR) of earlier time point to show any changes in glomerular filtration function in response to XO inhibitor (before and after). Histological analysis of whole kidney should be included along with glomerular scoring to show the overall damage and impact of XO inhibitor. All the mRNA analysis that is performed in whole kidney and not from the glomeruli. Although authors showed major injury/recovery in glomeruli and not from the whole kidney. It will be difficult to justify the impact of few glomeruli on overall kidney. Experiments from isolated glomeruli or more justifications are needed to convince the findings. Effect of oxypurinol, a permanent inactivator of xanthine oxidase should be discussed or experimentally proved that it can enhanced the DKD injury.

Author should explain more details about the limitations of the present study in discussion section.

Author Response

We thank the reviewer for the thoughtful comments.

>Author should measure the urine albumin creatinine ratio (UACR) of earlier time point to show any changes in glomerular filtration function in response to XO inhibitor (before and after).

Certainly, it may be better to examine the change amount before and after to see the effect. However, the collection of urine has a great influence on the subsequent growth, and it is feared that the effect of the model would be lost if KK-Ay mice, obese type 2 diabetes lose their weight. We thought it was not appropriate to increase the number of measurements. Therefore, the effect of the type 2 diabetes model was evaluated by preparing non-diabetic Control mice.

>Histological analysis of whole kidney should be included along with glomerular scoring to show the overall damage and impact of XO inhibitor. All the mRNA analysis that is performed in whole kidney and not from the glomeruli. Although authors showed major injury/recovery in glomeruli and not from the whole kidney.

We focused on the glomerular damage in this study, because we found less evident change in tubules and vascular than in glomeruli about anatomy, fibrosis, and/or cell invasion, although the findings in renal stroma might be important. About mRNA analysis, we discussed in limitation as below:

“All the mRNA analysis that is performed in whole kidney and not from the glomeruli, therefore the relationship between the gene expression and the effect of febuxostat on the glomerular protection might not be direct.”

>It will be difficult to justify the impact of few glomeruli on overall kidney.

We evaluated 40 glomeruli per one mouse, in a blinded manner on unidentified section as shown in Materials and Methods section. The estimated 95% confidence interval of glomerular score for one mouse is small enough (e.g.~±0.1). Therefore, we think the analysis is enough to be justified.

>Experiments from isolated glomeruli or more justifications are needed to convince the findings.

We are sorry for the lack of samples for the experiments from isolated glomeruli, but we found the obvious histological change in glomeruli, and added the sentences in the Discussion section as below.

“Since febuxostat administration to kk-ay mice in this study showed significant changes in glomerular lesions and no obvious changes in the renal tubules or mesenchyme. Therefore, we focused on the histological glomerular change, however, the inflammation and/or fibrosis in tubulointerstitium should be evaluated in the future study.”

>Effect of oxypurinol, a permanent inactivator of xanthine oxidase should be discussed or experimentally proved that it can enhanced the DKD injury.

We added sentences discussion section as below;

“The other xanthine oxidase inhibitors such as allopurinol, its metabolite oxypurinol, and topiroxostat should be evaluated in our model, to clarify the glomerular protective effect of febuxostat is class-effect.

>Author should explain more details about the limitations of the present study in Discussion section.

We explain our limitation as describe above.

Round 2

Reviewer 1 Report

Authors have addressed all points mentioned in the previous version of the article. The current form of review is well presented, and I have no additional comments.